# Reaction of Partially Methylated Polygalacturonic Acid with Iron(III) Chloride and Characterization of a New Mixed Chloride–Polygalacturonate Complex

**DOI:** 10.3390/molecules29040890

**Published:** 2024-02-17

**Authors:** László Kótai, Károly Lázár, László Ferenc Kiss, Klára Szentmihályi

**Affiliations:** 1HUN-REN Institute of Materials and Environmental Chemistry, Research Centre for Natural Sciences, Magyar Tudósok Körútja 2, H-1117 Budapest, Hungary; 2HUN-REN Centre for Energy Research, EKBI, Konkoly Thege Miklós Út, 29-33, H-1121 Budapest, Hungary; lazar.karoly@energia.mta.hu; 3HUN-REN Wigner Research Centre for Physics, H-1525 Budapest, Hungary; kiss.laszlo.ferenc@wigner.hu

**Keywords:** partially methylated polygalacturonic acid, iron–polygalacturonate anti-anemic agent, redox reaction, average molecular weight, double ion exchange, binuclear (hydr)oxo bridged iron centers

## Abstract

We have described a new route for the preparation of partially methylated polygalacturonic acid containing hydrolyzed (acidic) and unhydrolyzed (methyl esterified) carboxylate groups in a ratio of 1:1 (PGA, compound **1**), and one of its basic Fe^III^—salts (compound **2**) with a ~1:2 Fe^III^:GA stoichiometry (GA means galacturonic acid and methylated galacturonic acid units). The partially hydrolyzed pectin was transformed into compound **1** with the use of double ion exchange with a strongly acidic macroreticular sulfonated styrene–divinylbenzene copolymer as a hydrogen ion source. The reaction of compound **1** with FeCl_3_ resulted in compound **2**. Compound **2** has a polymeric nature and contains binuclear Fe^III^(µ-O)(µ-OH)Fe^III^ core units with two kinds of distorted octahedral iron geometries. The salt-forming acidic and methylated GA units of compound **1** are coordinated to Fe^III^ centers in asymmetric bidentate-chelating and -bridging (via C=O group and glycosidic oxygen) modes, respectively. Two kinds of outer-sphere chloride anions were also detected by XPS in various chemical environments fixed by different sets of hydrogen bonds. We also observed a partial reduction of Fe^III^ into Fe^II^ due to the ring-opening of the chain-end GA units of compound **1**. This reaction provides a new route to determine the number of chain-ends in compound **2**, and with the use of the number of GA units calculated from charge neutrality, the average length of these chains and the average molecular weight were also determined. The average molecular weight of the partially methylated polygalacturonic acid used in the industrial-scale production of commercial anti-anemic iron–polygalacturonate agents was ~50,000 g/mol. Compound **2** was also characterized by IR, Mössbauer, and X-ray photoelectron spectroscopy, and magnetic susceptibility measurements. These results on the structure and average molecular weight of basic iron(III) polygalacturonate provide a tool to design Fe-PGA complexes with tuned iron-releasing properties.

## 1. Introduction

Poly(galacturonic acids) and their various iron salts have great importance as anti-anemic agents added to various pharmaceutical compositions [1,2,3,4,5,6,7]. Their zinc, magnesium, and iron salts are also widely used in commercial products as food supplements (Ferrocomp^TM^, Multicomp^TM^, In Vitro Ltd., Dunakeszi, Hungary). The properties of these metal polygalacturonates depend on the method of preparation of poly(galacturonic acid) from pectin by alkaline, acidic, or enzymatic hydrolysis. Accordingly, the demethylation degree and polymer chain length (the average molecular weight and the distribution of the chain lengths) of polygalacturonic acids vary in a wide range, and are generally shorter than the chain lengths of the precursor pectin [8,9,10,11,12,13,14]. The biochemical properties of iron-complexed polygalacturonic acid derivatives—including the release and bioavailability of iron by the organs—strongly depend on the valence of iron, the strength and environment of the coordination around iron ions, and the solubility or reactivity of these polygalacturonate polymers. The main factors of solubility for these complexes, however, are the average molecular weight and polarity of the complexes. Since the composition, structure, and coordination characteristics of iron–polygalacturonate complexes have important roles in their stability, biological properties, solubility, and bioavailability, these basic chemical parameters should be known for the recently prepared new iron–polygalacturonate complexes as well.

The preparation of polygalacturonic acids as metal polygalacturonate precursors from their alkali salts with the addition of a mineral acid results in serious challenges due to the gel-like behavior of free polygalacturonic acids. Numerous instrumental methods have been developed to determine the average molecular weight of poly(galacturonic acids), including the use of light scattering [13,14], osmotic pressure or viscosity measurements [10,14], gel-permeation chromatography or size-exclusion chromatography [8,9,10,11,12,13], or combined methods such as size-exclusion chromatography coupled with multi-angle light-scattering photometry [14]. There has not been an easy and fast method that does not require precision analytical instruments to determine the molecular weight (*M*_w_) and polymerization degree (*n*) of polygalacturonic acids which can be transformed into metal salts.

Average molecular weight (M) has a great influence on the physiological properties of metal polygalacturonates. Therefore, the preparation of polygalacturonic acids with known average molecular weight is essential to prepare iron or other metal polygalacturonates. The most effective pharmaceutical compositions contain complexes which show similar iron bioavailability to that of the iron deposits in the livers of mammals (which are expected to be below 100,000 g/mol).

In this paper, we provide a new alternative preparation method for partially methylated polygalacturonic acid (PGA, compound **1**) from alkali-hydrolyzed pectin solutions with a double ion exchange in a porous diaphragm-separated ion-exchanger system and one of its basic iron(III) salts (compound **2**). The basic iron(III)–polygalacturonates were introduced as iron-supplement materials in pharmaceutical compositions, but there is no knowledge about either their average molecular weight or structural features. This salt was studied by Mössbauer, ESR, IR, and X-ray photoelectron spectroscopy, and magnetic susceptibility measurements, and an easy method is given to determine its average polymerization degree and the average molar weights of compounds **1** and **2**, as well as their structure and coordination.

## 2. Results and Discussion

### 2.1. The Preparation and Characterization of Compounds ***1*** and ***2***

#### 2.1.1. Hydrolysis of Pectin

The hydrolysis of pectin by the demethylation of ester groups in the presence of KOH or NaOH is a well-known method to prepare alkali metal polygalacturonates [15]. The pectin hydrolysis was tested with 0.5 M and 1 M NaOH at 70 °C and 25 °C for 40 min and 30 min, respectively. The solutions were acidified, and their carboxylic acid content was determined. The acid capacities of these samples were 3.8 and 4.7 mekv/g, respectively. Hydrothermal reactions performed with 1 M NaOH or 1 M HCl led to resinous material, whereas hydrolysis with water at 130 °C without the addition of alkali or acidic components resulted in polygalacturonic acid with an acid capacity of 4.0. 

#### 2.1.2. The Preparation of Compound **1** with the Use of Double Ion Exchange

The double-ion-exchange method used for the preparation of compound **1** is a process where two ion exchange reactions are performed at the same time (between a solid and a solution phase in each), in a way that the two solid phases are separated physically from each other, but the same solution phase is in contact with both solid phases. In order to reach this condition, the solution phase is divided into two parts with a porous diaphragm within the instrument, which ensures the diffusion of ions and solvent molecules between the two halves of the reactor but prevents the direct mixing of the two solid phases (Figure 1).

The double-ion-exchange procedure for the preparation of compound **1** consists of two different reactions. The first ion-exchange reaction is an exchange between the polygalacturonate-bound alkali ions and hydrogen ions:Sugar polymer—COONa + H^+^ = Sugar polymer—COOH + Na^+^

This is an equilibrium process, formally a reaction between a weak carboxylic acid sodium salt and a strong acid, in which the strong acid liberates the weak one from its salt. The second ion-exchange reaction is a reaction of a solid ion exchanger (sulfonated styrene–divinylbenzene copolymer) in its acidic form with the solution containing alkali-ions from the PGA salt:Polymer—SO_3_H + Na^+^ = Polymer—SO_3_Na + H^+^

A sulfonated ion-exchanger phase was selected as the solid acid. Since these solids cannot react with each other directly, these two reactions can only be performed in a U-tube reactor, in which these solids are contacted with the same solution. Although these two solid phases are separated with a diaphragm, the porous diaphragm ensures the ion diffusion processes via the common liquid medium (water). These two processes are formally combined into a reaction when the strong sulfonic acid liberates the weak polygalacturonic acid from its salt with the formation of the solid sodium salt of the strong acid.
Sugar polymer—COONa + Polymer—SO_3_H= Sugar polymer—COOH + Polymer—SO_3_Na

The sulfonated styrene–divinylbenzene copolymer swells easily and quickly and is strongly acidic. Thus, the driving force of the ion exchange—the liberation of the non-dissociated weak polygalacturonic acid—is strong. It is separated as a gel-like solid phase and is eliminated from the complex equilibrium. The reaction ends when the decrease in pH stops (Appendix A). Since both the demethylation and ion exchange rates are strongly temperature-dependent, we also studied the de-esterification and double-ion-exchange processes at 45–50 and 70–75 °C (Appendix A).

High-temperature (50 and 75 °C) demethylation with the subsequent double-ion-exchange treatment gave lower final pH values (2.04 and 1.88) and higher acid capacities (5.7 and 5.9 mekv/g), which were 12 and 16% higher for samples E and F, respectively, than those found for the commercial polygalacturonic acid.

#### 2.1.3. The Preparation of Compound **2**

Polygalacturonic acid easily reacts with iron(III) salts with the formation of iron(III)–polygalacturonate complexes with a ca. 1:2 Fe^III^:GA stoichiometry [16,17]. We reacted the partially methylated poly(galacturonic acid) prepared by us (compound **1**) with iron(III) chloride hexahydrate. The stoichiometry of the product (compound **2**) was GA/Fe = 2.11 (Appendix A), and due to the partial ester content, as the chemical analysis showed, chloride counter ions were also present, which compensate for the residual charges of Fe^III^ ions. The analysis data for our compound are given in Appendix A.

A PGA-Fe^III^ complex sample was prepared with the use of 50% excess FeCl_3_. The sample (compound **2***), however, contained inorganic basic iron(III) salts, as the Mössbauer studies showed (Appendix A); therefore, only compound **2** was studied further. The chloride ion content in compound **2** shows that the extra valence of iron(III) is neutralized by a chloride ion (the Fe:GA units’ stoichiometry is less than 1:3); furthermore, only half of the GA units contain ionic carboxylate, and the other part remains in methylated non-ionic COOMe groups. The gel was dried, and according to our DSC study (Appendix A), the light brown powder contained some adsorbed and chemically bound water, which was eliminated in consecutive endothermic processes at ~100 and ~150 °C, respectively.

### 2.2. Spectroscopic Characterization of Compound ***2***

The IR, Mössbauer, and ESR spectra of iron–polygalacturonates depend on their water content because, during drying, the water is eliminated and other oxygen-ligated coordinated environments can form [17,18]; therefore, we studied only samples dried at 105 °C for 2 h.

#### 2.2.1. Mössbauer Spectroscopic Results of Compound **2**

The Mössbauer study on compound **2** confirmed the appearance of a minor Fe^II^ environment due to the redox reaction of chain-end GA units, and unexpectedly showed the presence of two similar Fe^II^ and two similar Fe^III^ environments (Figure 2, Table 1).

The isomer shift values of each Fe^II^ and Fe^III^ environment in these compounds show a high-spin octahedral environment for every Fe^II^ and Fe^III^ ion.

Another PGA-Fe^III^ sample (compound **2***) was prepared in an analogous way but with a two-fold excess of FeCl_3_. Compound **2*** contained some ferromagnetic iron oxide particles as well; thus, further studies were completed only on compound **2**. Octahedral Fe^III^ coordination has a t_2g_^3^e_g_^2^ configuration; the orbitals are half-filled. Due to symmetric charge distribution, no quadruple splitting is expected. The QS 0.62 mm/s and 1.10 mm/s values for the (a) and (b) Fe^II^ environments strongly suggest that there are two differently distorted octahedral coordination Fe^III^ environments in compound **2**. The high-spin Fe^II^ (t_2g_^4^E_g_^2^ in an octahedral environment) has asymmetric electron distribution, causing strongly temperature-dependent QS values [19]. Low-temperature Mössbauer measurements were performed, which showed that the relative bonding strengths of the iron(II) and iron(III) ions to polygalacturonate oxygens were different (Table 1).

The distributions of each Fe^II^ or Fe^III^ species between the two locations marked with (a) and (b) in the polygalacturonate chain of compound **2** are 0.8 and 1.2, respectively. The coordination geometry is octahedral around both sites, but the quadruple splitting of d^5^ Fe^III^ ions shows a strong distortion of the octahedral geometry. The deviation from 1:1, as was expected from the dimeric nature of the iron-containing core, may be attributed to the atacticity of the PGA chain. This may result in two distorted bidentate-chelating or two bridging coordination scenarios for both iron centers of some Fe_2_(µ-O)(µ-OH) core units.

#### 2.2.2. XPS Results of Compound **2**

The XPS spectroscopy of compound **2** gave four Fe_2p_ signals (two 2p_3/2_ and 2p_1/2_ in each) and two Cl_2p_ signals (Appendix A, Appendix A). The identification of iron environments is difficult because energy values belonging to various Fe^II^ and Fe^III^ environments strongly coincide, so they cannot be assigned exactly. Therefore, we compared the XPS parameters of compound **2** with those of basic zinc and magnesium polygalacturonates (Appendix A and Appendix A). It can be concluded that the two well-separated Zn peaks belong to one environment—the peak at ~1046 eV is the 2p_1/2_ line of Zn—and there is also the usual doublet separation (23 eV) (~1022 is the 2p_3/2_ peak). Only one Mg environment was observed, and there was only one oxygen and two oxygen environments in the Zn and Mg compounds, respectively. This strongly suggests that the Zn and Mg compounds have similar structural motifs to other divalent metal polygalacturonates, whereas the high charge density of Fe^III^ ions in compound **2** results in other kinds of arrangements around the metal ions.

The carbon environments are similar for all studied compounds. The lowest, middle, and highest energy components belong to the ethereal and carboxylate carbons containing CH. The O1s peak at ~531.8 eV belongs to the ionic OH group bound to magnesium, where the other oxygens give a combined peak at ~532.9. This peak coincides with the peaks of other oxygens in the compound. The ionic OH peak of the Zn compound and the bridged OH peak in the XPS spectra of compound **2** completely coincide with other oxygen peaks of the polygalacturonate units. The formation of a bridge between the two metal ions might have a key role in the formation of the inequivalent metal-centered octahedral arrangements. In principle, the coordination of acidic or methylated carboxylate groups could make two kinds of coordination, but in this case, the zinc and magnesium salts would have two kinds of metal environments as well. Thus, the binuclear structural motifs bridged by (hydr)oxo groups might play an important role in the formation of diverse metal centers. The outer chloride ion compensates for the third charge of Fe^III^, and the presence of two kinds of chloride ions can be explained by the difference in their positions in the lattice; thus, they are probably fixed in different positions towards their environments by different hydrogen bonding sets.

#### 2.2.3. ESR Results of Compound **2**

The ESR spectrum of compound **2** shows isotropic *g* values around 4 (*g* = 6.19, 4.50, and 4.19) which belong to the distorted cubic symmetry of high-spin Fe^III^. Micera et al. [17] found a broad band at *g* = 4.3 and 2.0 in the ESR spectrum of an iron(III)–polygalacturonate, and these signals were assigned to isolated Fe^III^ ions in a rhombic crystal field and Fe^III^-O-Fe^III^ super-exchange interactions, respectively (Appendix A). This shows that hydration, as shown earlier [17,18,19], and the ratio of methylated carboxylate groups and synthesis conditions have an important role in the structure of iron(III)–polygalacturonate samples. For example, the drying conditions cause dehydration and have an influence on the structure and spectroscopic parameters of PGA-Fe^III^ samples, not only due to the elimination of water from the coordination sphere, but also due to the condensation of Fe^III^-OH groups into Fe(µ-O)Fe moieties. The ESR signals of high-spin Fe^II^ detected in the sample due to the reduction by the chain-end GA units by Mössbauer spectroscopy and XPS are expected to be very wide and weak at room temperature, and accordingly, we could not observe them in the room temperature ESR spectrum of compound **2**.

#### 2.2.4. Magnetic Measurements on Compound **2**

Measurements of the magnetization curves for compound **2** as a function of the magnetic field show relatively weak sensitivity regarding the *J* = (*L* + *S*) values (Figure 3). The magnetization as a function of temperature measured at *H* = 1 kOe shows a Curie–Weiss behavior with a Curie constant of *C* = 1.78 ± 0.1 cm^3^.K mol^−1^ and *θ* = –1.5 to –2 K, where the molecular weight is *w*_m_ = 507.5 g.mol^−1^ (Figure 4), which hints at the existence of magnetic interactions.

The effective magnetic moments calculated from the temperature dependence of *χ*_M_*T* (where *χ*_M_ = *Mw*_m_/*H* is the molar susceptibility) are µ_eff_ = 4.41 and 3.98 B.M at 298 and 100 K, respectively (Figure 5). These are lower than the theoretical values of isolated Fe^III^ (5.9 BM), but higher than the expected value for a low-spin Fe^III^ ion (3.87 B.M). This is very typical for Fe(µ-OH)Fe-type multinuclear coordination units [20], which suggests that our complex might contain similar coordination environments. Due to the Fe:GA = 1:2 stoichiometry, the 2 × 4 coordination sites of the octahedron/two irons are free for each binuclear unit, because 2 × 2 sites of the theoretical 2 × 6 sites are occupied by the bridging oxygen or hydroxy groups. These free sites might be filled with 2 × 2 = four GA units (Fe:GA = 1:2) or other ligands. Since every GA unit is coordinated via two donor atoms, this corresponds to four occupied coordination places across the two octahedra. The ligation of the residual four coordination sites may be fulfilled with the second (bidentate) coordination of the four coordinated GA units, or by donor groups other than GA units, e.g., water, a hydroxide ion, or a chloride ion. If we suppose that all GA units are bidentate, no chloride, hydroxy, or water ligand coordination is possible. 

Half of the carboxylic acid groups in our PGA are in COOH form, whereas the other half are in COOMe (esterified) form. The acidic COOH groups transform into ionized carboxylate groups in compound **2**. The coordination mode of the carboxylate ion might be monodentate, or two kinds of bidentate (chelating and bridging). The lower donor strength of the methylated carboxylate group prefers the monodentate coordination mode via the C=O groups. The GA units, however, have other O-donor atoms than the hydroxyl or glycosidic O atoms, which might make the GA units into bi-or polydentate ligands. The ligation of hydroxy groups was abandoned, but the coordination of glycosidic O atoms was found in transition metal polygalacturonates [21]. Therefore, we studied the IR spectrum of compound **2** in detail to clarify the coordination mode of GA units in this dried complex. 

#### 2.2.5. IR Spectroscopy of Compound **2**

The IR spectrum of compound **2** contains two partly covered ν_as_(C=O) mode bands at 1745 cm^−1^ and 1612 cm^−1^, belonging to the methylated and carboxylate groups, respectively (Appendix A). The analysis of the curve shows that there is a weak and a strong component at 1835 cm^−1^ and 1748 cm^−1^ for the ν_as_(C=O) of methylated carboxylate, respectively, whereas three bands at 1673 cm^−1^, 1616 cm^−1^ and 1567 cm^−1^ belong to the ionic carboxylate groups coordinated with Fe^III^. The shift of the methylated ν_as_(C=O) band of partly hydrolyzed polygalacturonic acid from 1742 cm^−1^ to 1745 cm^−1^ shows coordination between the Fe^III^ and carbonyl oxygen of the -C(= O)OMe groups, respectively. The difference between the wavenumbers of symmetric (ν_s_(C=O)) and antisymmetric (ν_as_(C=O)) modes of carboxylate groups is characteristic of their coordination mode. The Δ = ν_as_(C=O) − ν_s_(C=O) values of the corresponding antisymmetric/symmetric mode band pairs show unidentate (Δ = 410 cm^−1^, 359 cm^−1^) and asymmetric chelating (Δ = 311 cm^−1^, 304 cm^−1^, 266 cm^−1^) coordination modes of the methylated and acidic carboxylate ions in compound **2**, respectively. The changes in band positions of the carboxylate ion deformation (from 790 cm^−1^ to ~820 cm^−1^), ring-breathing (from 735 cm^−1^ to ~767 cm^−1^) and C(1)-O_glycoside_-C(4′) linkage-stretching modes (from 956 cm^−1^ to 961 cm^−1^) in the spectra of compounds **1** and **2**, respectively, show the carboxylate ion and glycosidic oxygen coordinations to iron. No coordination of sugar alcohol groups was observed.

The far-IR spectra of compound **2** (the Fe-Cl normal modes are expected to appear in the far-IR range) (Appendix A) did not show bands in the range expected for Fe-Cl bonds [22,23]. The similarity of the far-IR spectra of compound **2** and a mixed-valence Fe^II^,Fe^III^-PGA complex, prepared from iron(II) sulfate [16] in the absence of chloride ion, confirms the lack of any Fe-Cl band. The distorted octahedral coordination mode of iron in compound **2** and the low value of GA units/Fe ions in compound **2** (2.1) strongly suggest that the coordination mode of the GA units is bidentate. The strong and different distortion of Fe octahedra suggests different coordination modes of acidic and methylated GA units, which are asymmetric chelating bidentate and bridging bidentate modes (one coordination site is the oxygen of C=O group; the other is the glycosidic oxygen atom), respectively. No coordinated water or chloride ions were detected in compound **2**.

Based on the electroneutrality principle, if two iron ions are linked with two bridging oxygens in a four-membered cycle (Fe^III^(µ-O)_2_Fe^III^(2+) in compound **2**, and this dimer unit is coordinated with four GA groups (two ionic and two esterified, that is, two negative charges), an outer-sphere chloride ion should not be present, as was found by elemental analysis (Appendix A). Thus, one of the µ-O bridges is a µ-OH bridge, and the missing negative charge carrier is the outer-sphere non-coordinated chloride ion. The presence of a µ-Cl bridge or an outer-sphere OH ion can be shown on the basis of the far-IR data and the ionic nature (AgCl formation with AgNO_3_) of the chlorine content. The Fe^III^(µ-O)(µ-OH)Fe^III^(3+) core in compound **2** contains four coordination sites twice, and three charges. Two coordinated bidentate ionic carboxylates with two negative charges occupy four coordination sites of the Fe^III^(µ-O)(µ-OH)Fe^III^(3+) core, and the two bridging bidentate GA units (monodentate-methylated carboxylate and monodentate glycosidic oxygen coordination) occupy the remaining four sites. The differences between the two kinds of coordination places of GA units in compound **2** distort the octahedral geometry around Fe centers. Since Fe^II^ ions formed during the reduction can occupy the same sites in the PGA unit with the same octahedral geometry, two kinds of Fe^II^ environments also occur (see Mössbauer spectroscopy and XPS results).

These coordination elements (Figure 1) can be combined into egg-box-like structures [24] via the bridging of each chain by the coordination of the glycosidic O atoms and iron centers of another chain. The two kinds of each Fe^III^ and Fe^II^ coordination environment may be attributed to the presence of Fe centers coordinated with each kind of carboxylate/methylated carboxylate end group, whose coordination strengths are not the same. This results in different distortions and charge distributions around each Fe^II^ and each Fe^III^ center. The dependence of Mössbauer shifts of Fe^II^ and Fe^III^ centers on the hydration detected earlier [17]; however, our work shows that the methylation degree of PGA has enormously high importance in the existence of two different Fe^II^ and two different Fe^III^ environments. Furthermore, the presence of two kinds of Fe^II^ environment may also be attributed to the same coordination modes, and shows that the reduction of Fe^III^ into Fe^II^ is preceded by the complex formation of Fe^II^ with the sugar rings containing chains; thus, the coordination environments of Fe^II^ and Fe^III^ ions have competitive reactions.

Non-covalent interactions, e.g., hydrogen bonds of various OH groups, water, and oxygen atoms located in the structure, have an important role in the arrangement of chains and retaining water molecules in the samples. However, the influence of these non-covalent interactions on IR spectral characteristics could not be determined exactly due to the nature of O-H vibrational modes (intense and wide bands), which resulted in covering the various O-H group modes, resulting in very wide complex band systems (Appendix A).

### 2.3. Determining the Polymerization Degree (Average Molecular Weight) of Polygalacturonic Acid and Its Fe(III) Salt

Previous studies on iron(III)–polygalacturonate complexes showed that the iron(III)–polygalacturonate reaction products always contaminated a small amount of iron(II), indicating the existence of some redox interaction between the poly(galacturonic acid) and iron(III) ions as well [17,25]. The redox chemistry of monomer galacturonic acid is well established [26,27,28,29,30]; one D-galacturonic acid unit consumes four equivalents of Fe^III^ in an acidic pH environment with the formation of four equivalents of Fe^II^ and one equivalent of formic acid.

The aldehyde functional group (formed by the ring-opening of the cyclic form of galacturonic acid (GA)) was completely oxidized and eliminated as formic acid, and a new carboxylate group was built from the neighboring carbon atom (Figure 6) [26,27,28,29,30]. The acidic products ensure the optimal acidic pH environment (~3.5) for the redox reaction. The central iron(III) ion was coordinated with the carboxylate ions and the oxygen atoms of the five-membered ring, which has an essential role in the redox reaction due to the weakening of C-O bonds and its influence on the cyclic–linear aldehyde equilibrium.

In compounds **1** and **2**, the chemical character of the inter-chain (IC-GA) and the (last) chain-end GA units (LCE-GA) are different, independent of the methylation of the carboxylic groups of GA units. The aldehyde carbon and oxygen atoms of the IC-GA units are in strong C-O-C bonds, which connect the IC-GA units. Thus, the IC-GA rings cannot be opened, whereas the LCE-GA rings can be opened with the formation of free aldehyde functional groups. This aldehyde group reacts with Fe^III^ in the same way as in monomeric D-galacturonic acid. Accordingly, the number of PGA chains in a given amount of compound **2** is equal to the number of redox-active GA units (LCE-GA), which is proportional to the amount of the liberated Fe^II^ ions (Fe^II^/LCE-GA = 4). Thus, the number of LCE-GA units in compounds **1** and **2** can be determined by measuring the amount of liberated Fe^II^ ions.

If the amount of all GA units in a given amount of compound **1** or **2** (∑GA) is known, the average number of GA units per chain (*n*) can easily be calculated (*n* = LCE-GA/∑GA). To determine the overall amount of GA units in compounds **1** and **2**, we have to take into consideration the following:(1)Contrary to the free D-galacturonic acid Fe^III^ complex, compound **1** cannot completely neutralize all three charges of Fe^III^, and compound **2** that is formed contains a foreign counter ion (chloride due to the use of FeCl_3_) as well (Appendix A).(2)The non-demethylated GA units (GA-Me from the incomplete hydrolysis of pectin) in compound **1** do not provide any charge increment to neutralize the excess charge of Fe^III^ ions in the Fe^III^(µ-O)(µ-OH)Fe^III^ core.

If the overall Fe^III^ and chloride content of compound **2** are known, the amount of GA-H can be calculated with the use of the charge neutrality principle and the GA-H/GA-Me ratio. Since the Ga-H/Ga-Me ratio is the same in compound **2** as in compound **1**, the amount of overall GA units (GA-H + GA-Me) can also be calculated. The Fe^III^ content of compound **2** and the Ga-H/Ga-Me ratio in compound **1** were determined by iodometric and acidic–alkalimetric titrations [30,31], respectively. The (GA-H + Ga-Me)/Fe^III^ ratio was 2.12 (Appendix A).

To check this result, we completed an instrumental CHN elementary analysis to determine the overall number of GA-H and GA-Me rings. Taking into account the 1:1 GA-H and GA-Me content in PGA (the average carbon atom/ring value was 6.5), the carbon elemental analysis resulted in a (GA-H + GA-Me)/Fe ratio of 2.11, which is practically identical to the value obtained from the method of determining the amount of chloride (2.12). The (GA-H + GA-Me)/LCE-GA ratio gives the average polymerization degree, which was *n* = 255 and 254, derived from the determination of the total number of GA units with CHN analysis and our titrimetric method, respectively. Accordingly, the average molecular weights M_n_ (number-average) for compound **2** were 50,156 and 49,920 g/mol, respectively, which are very close to each other. The average molecular weight of commercial apple PGA was given to be between M_w_ = 47,000 and 55,000 g/mol by the producers [15]. The average molecular weight and polymerization degree of compound **1** were also determined by various instrumental methods (osmometric, light scattering, viscometry) [25]. The average molecular weight of compound **1** was between M_w_ = 47,000 and 49,000, which agrees well with the values given by our method (M_n_ = ~50,000) (Table 2).

The largest difference between the lowest and highest value of the polymerization degree of compound **1** determined by the known instrumental methods was 4.2%, whereas the smallest and largest differences between the values given by the different instrumental and (our) titrimetric methods were 2.4 and 6.2%, respectively (the average was 4.3%). 

## 3. Materials and Methods

Twice-purified Grinsted Wave 212 pectin (M_w_ = 40,000–60,000) was supplied by Danisco, Copenhagen, Denmark. All chemicals (36.5% aq. HCl, solid NaOH, FeCl_3_·6H_2_O, solid KMnO_4_, AgNO_3,_ solid KI, sodium thiosulfate pentahydrate, potassium chromate, 1% starch solution, and other analytical reagents) were supplied by Deuton-X Ltd. (Érd, Hungary). The pectin was hydrolyzed in the presence of 1 M HCl or 1 M NaOH at 25–80 °C and at 100–120 °C in an open system for a reaction time of 1–3 h. The hydrothermal treatment (autoclave experiment) was performed without adding chemicals at 100–120 °C for 1 and 2 h. 

The double-ion-exchange experiments were performed in a U-shaped tube consisting of a porous G4 glass diaphragm in the bottom part (Figure 1). The left branch of the tube was supplied with a pH meter and the volume of the alkali polygalacturonate solution was 50 mL. The right branch of the U-shaped tube was filled with water and an acidic form of a water-swelled solid Varion KSM (sulfonated, macroreticular) ion-exchange resin, Nitrokemia Co., Balatonfuzfo, Hungary. The amount of ion-exchanger material (100 g) was enough to ensure excess acid capacity relating to the amount of alkali ion content of the PGA salt. The double-ion-exchange processes were conducted until the pH stopped changing on the left side of the tube. The final pH values of the reaction mixtures are given in Appendix A. 

The ratio of the free acidic/methylated carboxylate groups in compound **1** was determined by acid–base titration with potentiometric end-point detection [23].

Compound **2** was prepared by adding 15.81 g iron(III) chloride hexahydrate dissolved in 1000 mL of water to compound **1** under vigorous stirring. The reddish-brown gel-like precipitate was stirred for a further 3 h and left to stand for 3 days. The gel was filtered off, washed twice with water, and then dried at 60 °C for 2 h. The iron(III) content of compound **2** was determined by iodometry [31]. The iron(II) content was determined by permanganometric titration according to the Zimmermann–Rheinhard method [25]. Chloride content was determined argentometrically [32]. For titrimetric purposes, 0.02 M KMnO_4_ and 0.1 M AgNO_3_ solutions were used. The calibrations of these solutions were performed with oxalic acid dihydrate in 10% aq. sulfuric acid solution, and with KCl standards, respectively. The endpoint of argentometric titration was detected by a potassium chromate indicator. The iron(III) content was measured by iodometric titration. Potassium iodide reacts with iron(III) with elementary iodine liberation, which was measured by sodium thiosulfate with a starch indicator.

The basic Zn and Mg polygalacturonates were prepared in an analogous way to the iron complex with the use of zinc and magnesium sulfates.

CHN analysis was performed with a Vario EL III. elemental analyzer (Elementar Analysensysteme GmbH, Hanau, Germany). Viscosity was measured with a micro-Ubbelohde viscosimeter in 0.005 M lithium oxalate/0.05 M lithium phosphate-buffered (pH = 7.30) solution. The constants of the Mark–Houwing equation ([η] = K·M^a^) were determined as *K* = 4.3691·10^−7^ and *a* = 1.8737 [25]. The light-scattering experiments were performed with a Sofica Ltd. (Malvern, UK) instrument in 0.05 M lithium phosphate-buffered solution containing 0.005 M lithium oxalate (pH = 7.30). The gel chromatography was carried out with a KhZh-1309 Laser micro-gel chromatograph, comprising a laser refractometric detector and Teflon micro-column filled with glycidyl polymethacrylate grains. The elution was performed with the lithium oxalate/lithium phosphate-buffered solution mentioned above. The details of these instrumental measurements are given in [25]. Further instrumental measurements were completed according to the descriptions given in our previous studies [33,34,35,36]. The solid-state IR spectra were recorded with the use of a BioRad-Digilab (Hercule, CA, USA) FTS-45-FT-IR spectrometer and a BioRad-Digilab FTS-40-FIR spectrometer for the 4000 ± 400 cm^−1^ and 400 ± 40 cm^−1^ range in KBr and polyethylene pellets, respectively [33,34]. DSC tests were performed on a Mettler Toledo (Columbus, OH, USA) TA4000 calorimeter between room temperature and 500 °C with heating rates of 10 °C/min under a N_2_ atmosphere [35]. The ESR spectra were recorded on a Jeol JES-FE/3X-ESR spectrometer (Jeol, Tokyo, Japan) upgraded for data acquisition (X-microwave (Austin, TX, USA) band, 100 kHz field modulation). The calibration for g-value measurements was performed with Mn^II^-doped MgO powder [34]. Mössbauer spectra were recorded at 300 and 77 K in constant acceleration mode with the KFKI standard spectrometer (^57^Co/Rh source) [36].

The magnetic measurements were performed with the help of a Quantum Design (Taipei City, Taiwan) MPMS-5S SQUID magnetometer at a temperature and field range of 5 to 300 K and 0 to 50 kOe using a gelatin capsule as a sample holder for the powder samples with a mass of 50 to 100 mg. During the measurements, the samples were surrounded by He as an exchange gas of several-millibar pressure.

## 4. Conclusions

(1)We developed a mild method to prepare polygalacturonic acid (compound **1**) with the partial de-esterification of pectin followed by a double-ion-exchange process with the use of a styrene–divinylbenzene copolymer-based sulfonated macroreticular ion-exchanger resin. The reaction of compound **1** with FeCl_3_ resulted in a basic PGA-iron(III) complex: compound **2**. The complex has a polymeric nature with ~1:2 Fe:GA stoichiometry and contains outer-sphere chloride ions.(2)Compound **2** contains two different and distorted Fe^III^ octahedral centers in Fe^III^(µ-O)(µ-OH)Fe^III^ units due to the asymmetrical ligation by two ionic carboxylates and two bidentate-bridging GA units containing methylated carboxylate with bidentate-chelating and C=O…Fe/glycosidic O…Fe coordination modes, respectively. The bridging ligands connect the neighboring Fe(µ-O)(µ-OH)Fe dimeric units into an egg-box-like polymeric structure. Two outer-sphere chloride anions are fixed in different environments by hydrogen bonds.(3)Fe^III^ was partially reduced into Fe^II^ in the reaction of compound **1** and FeCl_3_ due to the ring-opening of the chain-end galacturonic acid units of compound **1**. This reaction ensures an easy route to determine the number of polymer chains, the average polymerization degree, and accordingly, the average molecular weight of polygalacturonic acid (PGA) and its Fe^III^ salt (compounds **1** and **2**). The amount of Fe^II^ from the redox reaction of Fe^III^ and compound **1** is proportional to the total number of chain ends of polygalacturonic acid units. The ratio of overall galacturonic acid (free and methylated) and chain-end galacturonic acid gives the average polymerization degree of compounds **1** and **2**. The number of galacturonic acid units was determined both from CHN analysis and from Fe and chloride content with the use of the charge neutrality principle. The average degree of polymerization was *n* = 211 and 212 from the CHN analysis and titrimetric routes, respectively. The average molecular weight of the tested commercial polygalacturonic acid was ~50,000 g/mol.

## Data Availability

Data are contained within the article and Appendix A.

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
