# Peer review of "Reaction of Partially Methylated Polygalacturonic Acid with Iron(III) Chloride and Characterization of a New Mixed Chloride–Polygalacturonate Complex"

_molecules, 2024, doi:10.3390/molecules29040890_

Round 1

Reviewer 1 Report

Comments and Suggestions for Authors

In this manuscript, the authors describe the synthesis and characterization of partially methylated polygalacturonic acid (PGA, compound 1) and PGA-iron (III) complex (compound 2).

Both compounds were characterized by different analytical methods that thoroughly confirmed the molecular identity proposed by the authors.

However, various flaws are detected throughout the manuscript, as mentioned: 1)      On line 148, it is mentioned that DSC studies were carried out, but ESI Fig.1 corresponds to XPS analysis. 2)      On line 185, Table 2 is mentioned but not in the manuscript. 3)      In point 2.2.3., the ESR results are mentioned, but the spectra are not seen in the manuscript or the ESI. It is recommended that they show them. 4)      In the ESI Fig. 4, many spectra of many colors appear but do not indicate which one is each. It's hard to know how to correlate them. Labeling each one would be very helpful. 5)      It is advisable to include the far-IR spectra to corroborate the Fe-Cl vibrations (line 311) Given this, we believe that the authors should resolve these defects (minor corrections) to make the manuscript suitable for publication in this journal.

Author Response

We would like to express our many thanks for the reviewer's suggestions and their efforts to improve our manuscript. All the requested things have been done/answered, and the text was modified accordingly (marked by highlighting).  The detailed answers are below.

In this manuscript, the authors describe the synthesis and characterization of partially methylated polygalacturonic acid (PGA, compound 1) and PGA-iron (III) complex (compound 2). Both compounds were characterized by different analytical methods that thoroughly confirmed the molecular identity proposed by the authors.

However, various flaws are detected throughout the manuscript, as mentioned:

1)      On line 148, it is mentioned that DSC studies were carried out, but ESI Fig.1 corresponds to XPS analysis.

It has been revised ( The DSC curve was inserted into ESI)

2)      On line 185, Table 2 is mentioned but not in the manuscript.

It has been revised.

 3)      In point 2.2.3., the ESR results are mentioned, but the spectra are not seen in the manuscript or the ESI. It is recommended that they show them.

ESR spectrum has been inserted into ESI.

4)      In the ESI Fig. 4, many spectra of many colors appear but do not indicate which one is each. It's hard to know how to correlate them. Labeling each one would be very helpful.

The colors are used only to distinguish each band component (to shiow hiow many bands are in the spectra). Their positions were given in the Table (all band, all positions).  

5)      It is advisable to include the far-IR spectra to corroborate the Fe-Cl vibrations (line 311)

It has been inserted.

Given this, we believe that the authors should resolve these defects (minor corrections) to make the manuscript suitable for publication in this journal.

Reviewer 2 Report

Comments and Suggestions for Authors

Dear Authors,

After thoroughly examining your manuscript, I would like to express my highly positive feedback regarding your submission. Your study interests the readers, and the manuscript is well-organized and clearly -written. The aim of the study is precise, and the applied computational tools are adequate. I believe there is a room just for some symbolic technical improvements, so I request just some revisions based on the following remarks:

1.      Add some important outcome of study in abstract.

2.      The experimental and computational methods employed in this study should be discussed more detail. What concentrations were the solutions prepared?

3.      The authors must explain the effect of the NCI to studied properties.

4.      Why the authors limited the study of IR in the low range? They must show the effect of the NCI to the some modes.

5.      The quality of figure 2 is very bad, try to improve the quality of this figure.

6.      "Fig. x" should be written as "Fig. 7".

7.      List of rfes must be updated.

Based on these recommendations, we kindly advise the authors to address these issues and resubmit the manuscript for further consideration of possible publication.

Comments on the Quality of English Language

Acceptable

Author Response

We would like to express our many thanks for the reviewers suggestions and their efforts to improve our manuscript. All the requested things have been done/answered, and the text was modified accordingly (marked by highlighting).  The detailed answers are below.

After thoroughly examining your manuscript, I would like to express my highly positive feedback regarding your submission. Your study interests the readers, and the manuscript is well-organized and clearly -written. The aim of the study is precise, and the applied computational tools are adequate. I believe there is a room just for some symbolic technical improvements, so I request just some revisions based on the following remarks:

  1. Add some important outcome of study in abstract.

It has been done.

  1. The experimental and computational methods employed in this study should be discussed more detail. What concentrations were the solutions prepared?

These have been done. The missing concentrations have been given.

  1. The authors must explain the effect of the NCI to studied properties.

The non-covalent interactions -due to OH group in the bridge and a large number of hydroxy groups/oxygen atoms and water content of the samples have an important influence on the spectral properties, of course, however, because the large number of O-H bond species and the nature of O-H modes (wide and strong bands), these covers with each other, so unfortunately, we could not conclude detailed structural information from the positions of the combined wide complicated band systems. The text is modified accordingly.

  1. Why the authors limited the study of IR in the low range? They must show the effect of the NCI to the some modes.

The non-covalent interactions as hydrogen bonds influence could not be showed unambiguously -due to the covering of the similar hydrogen bound group’s bands resulting in a wide complex band system.  The complete IR spectrum has been inserted.

  1. The quality of figure 2 is very bad, try to improve the quality of this figure.

The quality of the picture goes wrong after inserting it into the word editor. The original good-quality one will be attached to the submission.

  1. "Fig. x" should be written as "Fig. 7".

The figure numberings has been revised.

  1. List of rfes must be updated.

The reference list has been updated.

Based on these recommendations, we kindly advise the authors to address these issues and resubmit the manuscript for further consideration of possible publication.

Reviewer 3 Report

Comments and Suggestions for Authors

The present paper deals with the novel route of preparation of Fe(III)polygalacturonate complexes with the well defined molecular weight of about 47-50000 g/mol (average polymeriyation degree 211). The complexes were prepared and characterized using Mossbauer, XRD, XPS, magnetic measurements and FTIR spectroscopy.

The topics is interesting and the methodology of characterization of complexes is sound, however the manuscript should be rewritten for better clarity and more interest for the readers.

I reccommend following corrections:

-Please emphasize more precisely the significance of the Fe(III)-PGA complexes for pharmaceutical applications in the Introduction section. Accordingly, is there an optimal molecular mass as the target of synthesis?

-Please define clearly the aim of the work at the end of the Introduction section. What is the target structure characteristics in light of potential applications?

-Please provide graphical representation of the prepared structures in the maintext - Results and discussion. Also, try to reduce the amount of structural data in the text, partially replace by graphical representations, and focus on the crucial results in the text.

-Please provide units in Table 3

-Please compare, in Results and Discussion or Conclusion sections, the obtained results with the available data from the literature on the similar compounds. What is novel and improved?

Author Response

We would like to express our many thanks for the reviewers suggestions and their efforts to improve our manuscript. All the requested things have been done/answered, and the text was modified accordingly (marked by highlighting).  The detailed answers are below.

The present paper deals with the novel route of preparation of Fe(III)polygalacturonate complexes with the well defined molecular weight of about 47-50000 g/mol (average polymeriyation degree 211). The complexes were prepared and characterized using Mossbauer, XRD, XPS, magnetic measurements and FTIR spectroscopy.The topics is interesting and the methodology of characterization of complexes is sound, however the manuscript should be rewritten for better clarity and more interest for the readers.

I reccommend following corrections:

-Please emphasize more precisely the significance of the Fe(III)-PGA complexes for pharmaceutical applications in the Introduction section. Accordingly, is there an optimal molecular mass as the target of synthesis?

The introduction has been revised.  The optimal average molecular weight is expected to be below 100 000 in pharmaceutically active compositions. We found that these treatments gave an advantageous molecular mass range ( ~50 000).

-Please define clearly the aim of the work at the end of the Introduction section. What is the target structure characteristics in light of potential applications?

The introduction has been improved according to the remark.

-Please provide graphical representation of the prepared structures in the maintext - Results and discussion. Also, try to reduce the amount of structural data in the text, partially replace by graphical representations, and focus on the crucial results in the text.

It has been done.

-Please provide units in Table 3

It has been done.

-Please compare, in Results and Discussion or Conclusion sections, the obtained results with the available data from the literature on the similar compounds. What is novel and improved?

It has been done.

Round 2

Reviewer 3 Report

Comments and Suggestions for Authors

The manuscript is now significantly improved and much clearer to read. There is just a minor remark - the end of last sentence in the Introduction is missing.